# Soil Collected in the Great Smoky Mountains National Park Yielded a Novel *Listeria sensu stricto* Species, *L. swaminathanii*

Catharine R. Carlin,[a] Jingqiu Liao,[a,b*] Lauren K. Hudson,[c] Tracey L. Peters,[c] Thomas G. Denes,[c] Renato H. Orsi,[a] Xiaodong Guo,[a] Martin Wiedmann[a]

aDepartment of Food Science, Cornell University, Ithaca, New York, USA
bDepartment of Microbiology, Cornell University, Ithaca, New York, USA
cDepartment of Food Science, University of Tennessee, Knoxville, Tennessee, USA

**ABSTRACT** Soil samples collected in the Great Smoky Mountains National Park yielded a *Listeria* isolate that could not be classified to the species level. Whole-genome sequence-based average nucleotide identity BLAST and *in silico* DNA-DNA Hybridization analyses confirmed this isolate to be a novel *Listeria sensu stricto* species with the highest similarity to *L. marthii* (ANI = 93.9%, isDDH = 55.9%). Additional whole-genome-based analysis using the Genome Taxonomy Database Toolkit further supported delineation as a novel *Listeria sensu stricto* species, as this tool failed to assign a species identification. Phenotypic and genotypic characterization results indicate that this species is nonpathogenic. Specifically, the novel *Listeria* species described here is phenotypically (i) nonhemolytic and (ii) negative for phosphatidylinositol-specific phospholipase C activity; the draft genome lacks all virulence genes found in the *Listeria* pathogenicity islands 1, 2, 3, and 4 as well as the internalin genes *inlA* and *inlB*. While the type strain contains an apparently intact catalase gene (*kat*), this strain is phenotypically catalase-negative (an unusual characteristic for *Listeria sensu stricto* species). Additional analyses identified a nonsynonymous mutation in a conserved codon of *kat* that is likely linked to the catalase-negative phenotype. Rapid species identification systems, including two biochemical and one matrix-assisted laser desorption/ionization, misidentified this novel species as either *L. monocytogenes, L. innocua,* or *L. marthii*. We propose the name *L. swaminathanii,* and the type strain is FSL L7-0020[T] (=ATCC TSD-239[T]).

**IMPORTANCE** *L. swaminathanii* is a novel *sensu stricto* species that originated from a US National Park and it will be the first *Listeria* identified to date without official standing in the nomenclature. Validation was impeded by the National Park's requirements for strain access, ultimately deemed too restrictive by the International Committee on Systematics of Prokaryotes. However, lack of valid status should not detract from the significance of adding a novel species to the *Listeria sensu stricto* clade. Notably, detection of non-*monocytogenes sensu stricto* species in a food processing environment indicate conditions that could facilitate the presence of the pathogen *L. monocytogenes*. If isolated, our data show a potential for *L. swaminathanii* to be misidentified as another *sensu stricto,* notably *L. monocytogenes*. Therefore, developers of *Listeria* spp. detection and identification methods, who historically only include validly published species in their validation studies, should include *L. swaminathanii* to ensure accurate results.

**KEYWORDS** *Listeria sensu stricto*, novel species, average nucleotide identity, *in silico* DNA-DNA hybridization, US National Parks, valid publication

Address correspondence to Martin Wiedmann, martin.wiedmann@cornell.edu.

*Present address: Jingqiu Liao, Department of Systems Biology, Columbia University, New York, New York, USA.

The authors declare no conflict of interest.

The identification of *L. swaminathanii* brings the total number of *Listeria* species to 27 as of April 15, 2022. For 58 years, the *Listeria* genus contained only six species (*L. monocytogenes*, *L. innocua*, *L. ivanovii*, *L. seeligeri*, *L. welshimeri*, and *L. grayi*) that were described between 1926 and 1984 (1–5). Beginning in 2010 with the identification of *L. marthii* (6) and *L. rocourtiae* (7), this genus saw a rapid expansion with a total of 11

species added between 2010 and 2015; in addition to *L. marthii* and *L. rocourtiae*, *L. fleischmannii* (8, 9), *L. weihenstephanensis* (10), *L. aquatica* (11), *L. cornellensis* (11), *L. floridensis* (11), *L. grandensis* (11), *L. riparia* (11), *L. booriae* (12), and *L. newyorkensis* (12) were added during this period. The 11 newly classified species considerably changed the taxonomy of the genus, notably 10 of these species lacked characteristics historically expected of *Listeria* (e.g., motility, growth at 4°C [13]); this expanded diversity led to a subdivision into two clades, designated *sensu stricto* and *sensu lato*, based on relatedness to *L. monocytogenes* (14, 15). The *sensu lato* clade is represented by the species showing a more distant relation to *L. monocytogenes*; this clade contains *L. grayi* as well as 10 of the 11 species described between 2010 and 2015. From 2018 to 2020, the *sensu lato* clade continued to expand with the addition of four novel species [*L. costaricensis* - 2018 (16), *L. goaensis* - 2018 (17), *L. thailandensis* – 2019, (18), and *L. valentina* – 2020 (19)]. Between 2010 and 2020, only one species, *L. marthii*, was added to the *sensu stricto* clade; this clade contains *L. monocytogenes* and those species most phylogenetically related to *L. monocytogenes* (*L. innocua*, *L. ivanovii*, *L. seeligeri*, *L. welshimeri*, and *L. marthii* as of 2020). By 2020, there were 15 novel species ($n = 1$ *sensu stricto*, $n = 14$ *sensu lato*) bringing the total number of validly published *Listeria* species to 21.

Identification and characterization of novel *Listeria sensu stricto* species is important to food safety as *L. monocytogenes* is a key member of this clade and the causative agent of listeriosis, a rare but severe foodborne disease (20). Specifically, testing for *Listeria sensu stricto* species is used to identify environmental conditions in food processing plants that indicate an increased risk for *L. monocytogenes* contamination, since *Listeria sensu stricto* species (i) grow under similar environmental conditions as *L. monocytogenes* (e.g., refrigeration temperatures [13]), and (ii) are frequently isolated from environments where *L. monocytogenes* is also detected (21–24). In 2021, as part of a project to characterize the prevalence of *Listeria* in soil throughout the contiguous United States (25), five novel *Listeria* species were identified, including four novel *Listeria sensu stricto* species, marking the first expansion of this clade since 2009. While there was sufficient scientific evidence for all five species to be classified as a novel *Listeria* species, only four of the five species (*L. cossartiae*, *L. farberi*, *L. immobilis*, *L. portnoyi*, and *L. rustica*) met the criteria to obtain valid standing in the nomenclature and were hence validly published (26). The fifth species (one of the four *sensu stricto*), described here and given the name *L. swaminathanii*, originated from soil collected in the Great Smoky Mountains National Park (GSMNP). Briefly, *L. swaminathanii* could not be validated because we were unable to obtain approved culture collection certificates, which is a prerequisite for valid publication (27). According to the International Committee on Systematics of Prokaryotes (ICSP; [28]), the US National Park's requirements for obtaining access to the type strain violate the International Code of Nomenclature of Prokaryotes (ICNP; [27]) policy for open access. Specifically, while the *L. swaminathanii* type strain is available from the American Type Culture Collection (ATCC), the US National Park's Material Transfer Agreement (MTA) associated with this strain has been deemed too restrictive to allow for recognition of *L. swaminathanii* as a new species; the same fate would occur regardless of which culture collection the strain is deposited. At the time of this writing, researchers from the University of Tennessee also isolated multiple *Listeria* species ($n = 5$) from GSMNP, including *L. monocytogenes*, *L. marthii*, *L. booriae*, the recently described *L. cossartiae*, and two strains that could not be classified to the species level (29), further illustrating the negative impact of existing access requirements for type strains. Novel species isolated in India face similar challenges as the Indian government also imposes restricted access to cultures (30), hence any species isolated in India cannot be validly published either. Thus, rules intended to protect the rights of discoveries and provide open access to the research community can, in some cases, create barriers to the formal validation of a new species. In the case of *L. swaminathanii*, this may result in a potential *L. monocytogenes* indicator organism from being excluded from *Listeria* spp. method validation studies due to confusion around the status of this species.

## RESULTS

**A soil sample from the Great Smoky Mountains National Park yielded *Listeria* isolates that could not be identified to the species level.** The novel species described here was isolated from soil collected in the Great Smoky Mountains National Park in NC, USA (Latitude 35.4726543, Longitude −83.851303). A total of 31 *Listeria*-like colonies were isolated from five soil samples that together yielded six different *sigB* allelic types (AT) representing three previously described species, including (i) *L. monocytogenes* (1 AT), (ii) *L. innocua* (1 AT), and (iii) *L. booriae* (3 ATs) along with one isolate that could not be classified to the species level (1 AT). The putative novel species is represented by five colonies that all generated the same, novel *sigB* AT (AT 166); these five isolates were designated FSL L7-0020$^T$, FSL L7-0021, FSL L7-022, FSL L7-0023, and FSL L7-0024. The observation that the *sigB* AT for these five isolates differed by 8 SNPs from the most closely related *sigB* AT (*L. marthii* AT 42), suggested that these isolates may represent a novel species.

**Whole-genome sequence-based phylogenetic analyses established *L. swaminathanii* is a novel *Listeria sensu stricto* species.** To determine whether the five isolates with *sigB* AT 166 represented a novel species, isolate FSL L7-0020$^T$ was designated the type strain with the proposed name *L. swaminathanii* and selected for whole-genome sequencing (WGS) followed by whole-genome-based species delineation assessment via (i) average nucleotide identity using BLAST (ANIb) (31), (ii) *in silico* DNA-DNA Hybridization (isDDH) (32), and (iii) the Genome Taxonomy Database Toolkit (GTDB-Tk) (33–35). The draft genome for *L. swaminathanii* FSL L7-0020$^T$ (GenBank accession number: JAATOD000000000) contained 13 contigs and had an $N_{50}$ length of 1,428,095 bp, an average coverage of 127×, a total length of 2.8 Mb, and G+C content of 38.6 mol%. The total length and G+C content are consistent with the range for current *Listeria sensu stricto* species genomes (2.8 to 3.2 Mb and 34.6 to 41.6 mol%, respectively) (13, 14). The parameters of this draft genome all met the recommended values for taxonomic evaluation set forth by Chun et al. (36).

WGS-based ANIb analysis revealed that *L. swaminathanii* FSL L7-0020$^T$ clustered with the *Listeria sensu stricto* clade and showed the highest similarity to *L. marthii* with an ANIb value of 93.9% (Fig. 1), which is below the 95% cutoff for species delineation (37). Analysis by WGS-based isDDH also yielded a value below the cutoff for species delineation (<70%) (37). Specifically, *L. swaminathanii* and the most similar reference genome (*L. marthii* FSL S4-120$^T$) yielded an isDDH value of 55.9% (confidence interval 53.1 to 58.6%). Additionally, GTDB-Tk failed to yield a species classification for the *L. swaminathanii* draft genome but did identify *L. marthii* as the most similar genome (FastANI value of 94.4%, AF value of 0.93); the taxonomy of all 34 reference genomes included in the analysis were correctly identified. The phylogenetic tree inferred from the GTDB-Tk output (Fig. 2) positioned *L. swaminathanii* among the *Listeria sensu stricto* clade where it clusters with *L. marthii* and *L. cossartiae*.

**L. swaminathanii yielded colony morphologies typical of nonpathogenic *Listeria* sp.** Following streaking of an overnight Brain Heart Infusion (BHI, Becton Dickinson) broth culture onto Modified Oxford agar (MOX, Becton Dickinson) and Listeria monocytogenes chromogenic plating medium (LMCPM, R&F Laboratories) agars, *L. swaminathanii* yielded colonies typical of *Listeria* species (38). Notably, the morphology exhibited by *L. swaminathanii* on both MOX and LMCPM was indistinguishable from what is expected of the other non-*monocytogenes sensu stricto* species (38). When grown on MOX, *L. swaminathanii* FSL L7-0020$^T$ yielded black colonies indicative of esculin hydrolysis that were round, had sunken centers, and a black halo; this morphology matches the current description for "typical" *Listeria* spp. growth on MOX (38). Phosphatidylinositol-specific phospholipase C (PI-PLC) activity is a virulence factor presently associated with the pathogenic species *L. monocytogenes* and *L. ivanovii*. On LMCPM agar, PI-PLC activity is generally detected by the chromogen X-inositol phosphate; colonies positive for PI-PLC activity appear blue-green, negative colonies are white (39). When streaked to LMCPM, *L. swaminathanii* FSL L7-0020$^T$ yielded colony morphologies consistent with *Listeria* spp. that are negative for PI-PLC activity. Specifically, *L. swaminathanii* yielded small, round, white colonies on LMCPM. *L. monocytogenes* 10403S generated blue-green colonies indicative of PI-PLC activity.

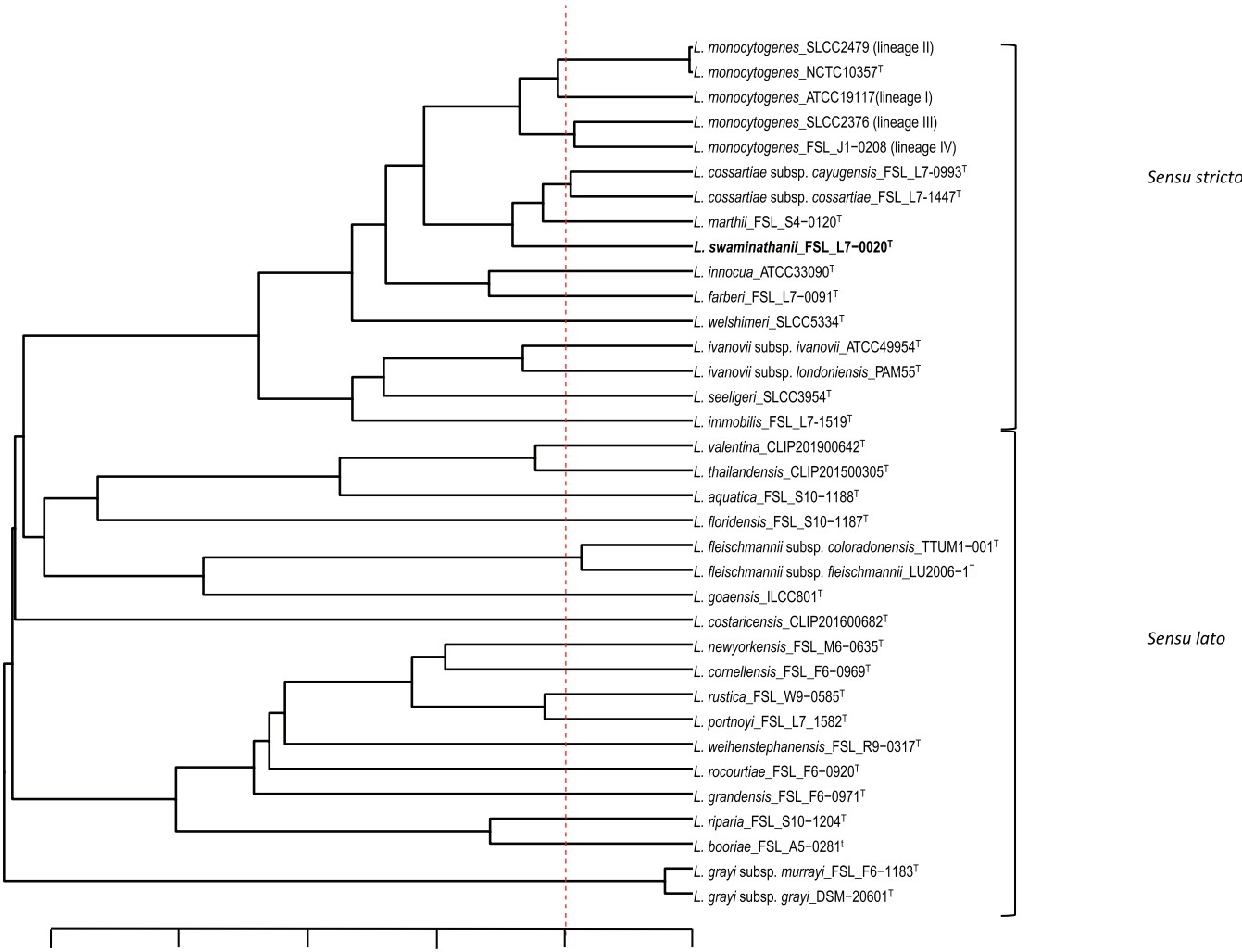

**FIG 1** UPGMA dendrogram based on Average Nucleotide Identity BLAST (ANIb) analysis of 34 reference genomes (consisting of the 30 *Listeria* species and subspecies type strains described as of June 11, 2021, and one genome representing each of the four *L. monocytogenes* lineages) and the *L. swaminathanii* FSL L7-0020[T] draft genome. The vertical red dotted line is placed at 95%, representing the species cutoff. The horizontal scale bar indicates ANI percentage similarity.

**Except for the catalase negative reaction, *L. swaminathanii* generated the expected biochemical results of a nonpathogenic *Listeria sensu stricto* species.** The standard *Listeria* reference method characterization tests we performed included (i) catalase, (ii) oxidase, (iii) Gram staining, (iv) beta-hemolysis on blood agar, (v) nitrate and nitrite reduction, and (vi) motility. Interestingly, the *L. swaminathanii* type strain FSL L7-0020[T] was catalase-negative; a characteristic not previously observed with any *sensu stricto* species (6, 13, 26, 38); however, several catalase-negative *L. monocytogenes* strains have been reported (40–43). Other than *L. swaminathanii* FSL L7-0020[T], the only other catalase-negative species reported to date is the recently described *sensu lato* species, *L. costaricensis* (16). Among the catalase-negative *L. monocytogenes* referenced above, one isolate had the catalase activity restored upon subculturing (42). However, when we subcultured *L. swaminathanii* FSL L7-0020[T], the isolate remained catalase-negative. Specifically, an additional biological replicate of FSL L7-0020[T] was subcultured to BHI, by selecting colonies grown on BHI agar and streaking them on a second BHI agar, along with four additional *L. swaminathanii* strains (FSL L7-0021, FSL L7-022, FSL L7-0023, and FSL L7-0024). The five *L. swaminathanii* strains share identical *sigB* ATs, and they all (including the type strain) retained a catalase-negative phenotype after subculturing. To further assess the absence of catalase activity, analysis of the draft genome

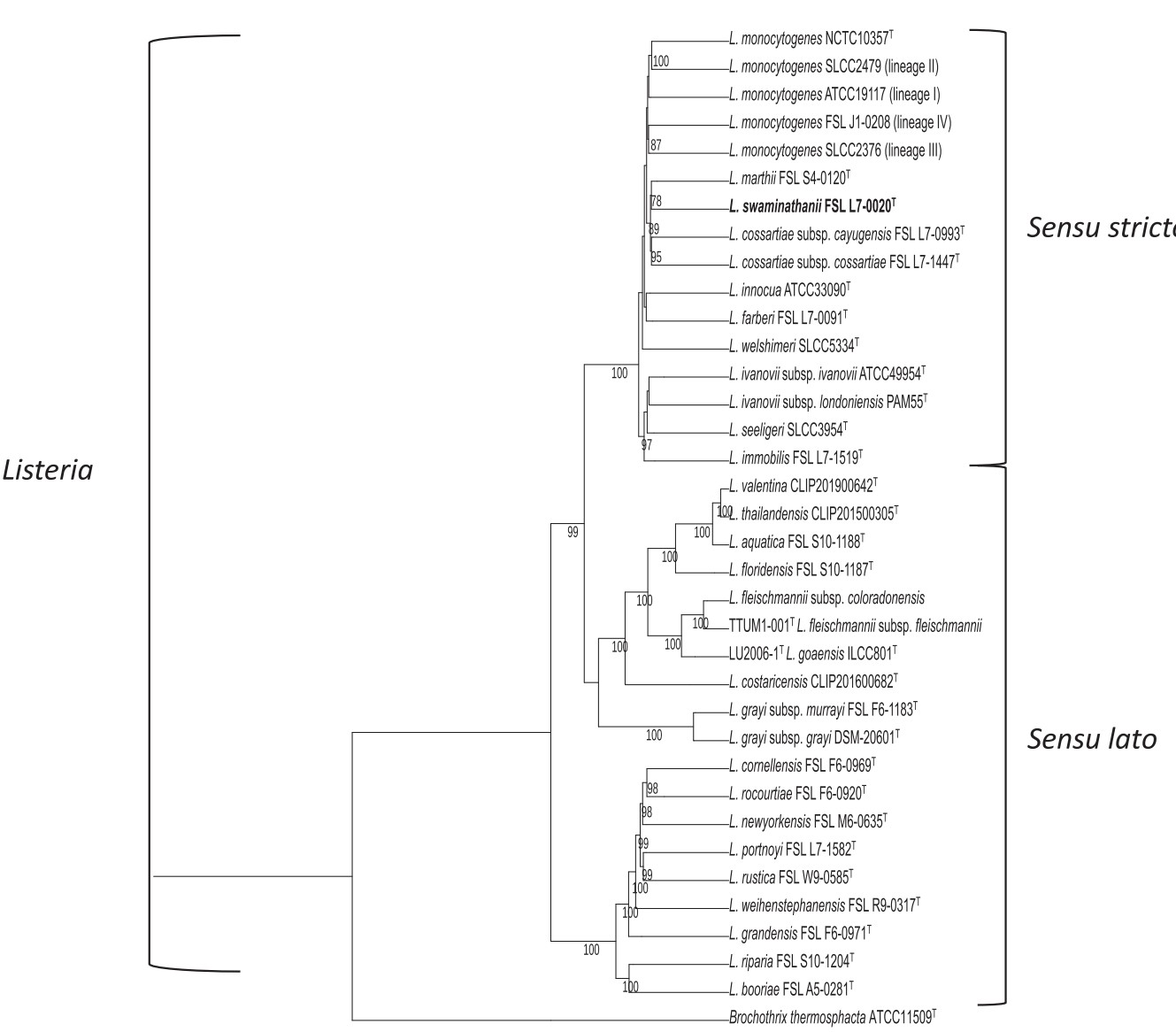

**FIG 2** Maximum Likelihood (ML) phylogenetic tree based on the GTDB-Tk analysis of 120 concatenated protein amino acid sequences of the same 34 reference genomes used for ANIb analysis and the *L. swaminathanii* draft genome. The phylogeny was inferred using RAxML v8.2.12 (62), and the best fit model for protein evolution, PROTGAMMAILGF, was determined using ProtTest 3.4.2 (70). The values mapped to the nodes represent bootstrap values based on 1,000 replicates; values <70% are not shown. The tree is rooted at the midpoint and includes the outgroup *Brochothrix thermosphacta* ATCC 11509[T].

for the *kat* gene was performed (see below for results). Other than the catalase reaction, the oxidase and Gram-stain results were consistent with what is currently expected for *Listeria* spp. (13). Specifically, the *L. swaminathanii* type strain FSL L7-0020[T] presented as an oxidase-negative, Gram-positive short rod. Sheep's Blood Agar (SBA, Becton, Dickinson) was used for hemolysis testing. Only *L. monocytogenes* 10403S lysed the red blood cells in the agar resulting in a clear zone of beta-hemolysis (a phenotype associated with *Listeria* pathogenicity); hence, *L. swaminathanii* is nonhemolytic. The absence of hemolysis is further supported by the absence of the hemolysin gene (*hly*) from the *L. swaminathanii* draft genome (described below).

None of the *Listeria* species described to date reduce nitrite, while nitrate reduction is currently only observed with the recently described *sensu lato* species (14, 16–19, 26). After the *L. swaminathanii* FSL L7-0020[T] nitrate broth enrichment was combined with Sulfanilic acid and N, N-Dimethyl-*a*-nathylamine, no red color change was observed until the

addition of zinc; a red color change was generated when these reagents were combined with the nitrite broth enrichment, indicating this species does not reduce nitrate or nitrite. The control strains performed as expected. Specifically, *L. monocytogenes* 10403S did not reduce nitrate or nitrite and *L. booriae* FSL A5-0281$^T$ only reduced nitrate.

Motility was assessed both microscopically and following stab inoculation into Motility Test Medium (MTM, Becton, Dickinson). For the microscopic method, wet mounts were prepared from BHI agar cultures grown at 25°C and 37°C for 24 h. Motility testing using MTM was performed by stab-inoculating the medium (purchased premade in 10 mL screw-cap tubes) with an isolated colony selected from BHI agar followed by incubation at 25°C with observations every 24 h for 7 days. *L. swaminathanii* FSL L7-0020$^T$, along with the *L. monocytogenes* positive control, exhibited motility at 25°C with both motility test methods; a tumbling movement was observed microscopically, and an umbrella-like growth pattern was observed following incubation in MTM agar. *L. swaminathanii* FSL L7-0020$^T$, along with both control strains, were nonmotile at 37°C. To date, *L. costaricensis* is the only *Listeria* species reported to be motile at 37°C (16), and *L. immobilis* is the only *sensu stricto* species that lacks motility at 25°C (26).

**The growth range and optimal growth temperature of *L. swaminathanii* is consistent with what is currently expected of *Listeria* spp.** The expected growth range for *Listeria* is currently listed as 0 to 45°C (13), although exceptions have been identified with several recently described *sensu lato* species that exhibit a narrower temperature range for growth, including eight *sensu lato* species that do not growth at 4°C (14, 16–19) and four species that do not grow at 41°C (7, 10, 26). Presently, all species grow optimally at either 30 or 37°C (14, 16–19, 26). *L. swaminathanii* generated growth at all temperatures tested. The least growth (4.34 log$_{10}$) was recorded after 10 days of incubation at 4°C, and optimal growth was achieved at both 30 and 37°C (9.28 and 9.30 log$_{10}$) after 24 h of incubation. *L. swaminathanii* along with the *L. monocytogenes* 10403S control strain both grew anaerobically. Detailed growth data can be found in Table S1.

**API *Listeria* analysis misidentified *L. swaminathanii* as *L. monocytogenes*.** *L. swaminathanii* yielded the numeric code 6110, which the apiweb database (bioMérieux V2.0, apiweb version 1.4.0) reported as "very good identification to the genus" with an 80% ID to *L. monocytogenes* and a T value of 0.62. Possible T values range from 0 to 1.0; the closer the value is to 1.0, the closer the biochemical test results are to what is considered "typical" for the species (44). *L. monocytogenes* was reported as the most likely species due to a negative result for the D-arylamidase activity, referred to as the DIM test (Differentiation of *innocua* and *monocytogenes*) (38). The discordant result leading to a T value of 0.62 is attributed to the negative result for rhamnose fermentation generated by *L. swaminathanii* FSL L7-0020$^T$. Differentiation from *L. monocytogenes* may be achieved via a negative hemolysis test. The same numeric code (6110) has also been reported for *L. marthii* (6) and *L. cossartiae* subsp. *cossartiae* (26). Phenotypically, *L. swaminathanii* FSL L7-0020$^T$ is most easily differentiated from *L. marthii* and *L. cossartiae* subsp. *cossartiae* by the lack of catalase activity. However, the catalase-negative phenotype appears to be a variable trait as the recent characterization of two additional *L. swaminathanii* strains by Hudson et al. showed that these two strains are catalase-positive (45). Further differentiating characteristics were determined following the API CH50 analyses described below. *L. monocytogenes* 10403S and *L. innocua* ATCC 33090$^T$ (the strain recommended by the manufacturer to verify the performance of the DIM) were tested to verify the API *Listeria* kit performance and generated the expected results for typical strains (numeric codes 6510 and 7510, respectively).

**API 20E results are consistent with classification of *L. swaminathanii* into *Listeria sensu stricto* and API CH50 results allow for further differentiation of *L. swaminathanii* from *L. marthii* and *L. cossartiae*.** The API 20E was utilized to perform a number of biochemical tests (i.e., Voges-Proskauer, indole utilization, urease activity, H$_2$S production) typically performed for *Listeria* characterization. *L. swaminathanii* FSL L7-0020$^T$ tested positive for Voges-Proskauer and negative for indole, urease, and H$_2$S production via the API 20E test, which is consistent with what is currently expected

of *Listeria sensu stricto* species. Specifically, all currently described *sensu stricto* species are Voges-Proskauer negative while the majority of *sensu lato* species (12 out of 15) are positive. Test results from the API CH50 identified a number of phenotypic differences between *L. swaminathanii* FSL L7-0020^T and both *L. marthii* and *L. cossartiae* (in addition to the unique catalase negative phenotype of *L. swaminathanii* FSL L7-0020^T detailed above). Specifically, *L. swaminathanii* FSL L7-0020^T is negative for fermentation of D-turanose and positive for glycerol and starch utilization while *L. marthii* ferments D-turanose and does not utilize glycerol. Although the starch result is not commonly used to differentiate *Listeria* and therefore often not reported, we found that the *L. swaminathanii* FSL L7-0020^T's ability to utilize starch differentiated this isolate from *L. cossartiae* (Tables 1 and S1). A summary of the results commonly reported for *Listeria* are presented in Table 1; additional API CH50 results are provided in Table S2.

**Three automated rapid identification systems (biochemical and matrix-assisted laser desorption/ionization-time of flight [MALDI-TOF] based) misidentified the novel *Listeria sensu stricto* species *L. swaminathanii* as well as the other recently reported *L. cossartiae*, *L. farberi*, and *L. immobilis*.** Vitek 2 (a biochemical rapid identification system) yielded a "good identification" for *L. swaminathanii* FSL L7-0020^T with a 91% probability of being *L. innocua*. Vitek MS V3.2 (a MALDI-TOF-based system) on the other hand identified *L. swaminathanii* FSL L7-0020^T as *L. marthii* with a confidence value of 99.9%. Previous novel species publications using Vitek 2, Vitek MS or Bruker's MALDI Biotyper (MBT) to characterize novel species did not yield a species identification; however, all these isolates were novel *sensu lato Listeria* species (17, 18). Our data suggest that, unlike novel *sensu lato*, novel *sensu stricto* species could be misidentified given their genetic and phenotypic similarities to the species currently represented in the respective databases. The Vitek 2 database contains strains representing six *Listeria* species (*L. monocytogenes*, *L. innocua*, *L. ivanovii*, *L. seeligeri*, *L. welshimeri*, *L. grayi*), and the Vitek MS database contains strains representing seven species, the same set as Vitek 2 plus *L. marthii*. We thus used both the Vitek 2 and Vitek MS systems to also screen additional *sensu stricto* species not included in these databases, including *L. farberi* FSL L7-0091^T, *L. immobilis* FSL L7-1519^T, and *L. cossartiae* (both subspecies *cossartiae* FSL L7-1447^T and *cayugensis* FSL L7-0993^T). For *L. farberi*, Vitek 2 yielded a low discrimination result with the potential to be *L. innocua*, *L. monocytogenes*, or *L. welshimeri*; the systems software recommended beta-hemolysis, CAMP, and xylose fermentation testing to discriminate the species identification further. *L. farberi* FSL L7-0091^T was identified as *L. innocua* with Vitek MS (confidence value 99.9%). *L. immobilis* FSL L7-1519^T yielded an excellent identification as *L. ivanovii* with Vitek 2 and was identified as *L. monocytogenes* (confidence value 99.7%) with Vitek MS. *L. cossartiae* subsp. *cossartiae* FSL L7-1447^T yielded the same identification reported for *L. swaminathanii* FSL L7-0020^T (*L. innocua* with Vitek 2 and *L. marthii* with Vitek MS). *L. cossartiae* subsp. *cayugensis* FSL L7-0993^T gave a low discrimination result with Vitek 2 and the possibility of being *L. innocua* or *L. grayi* due to the positive result for ribose fermentation seen with this strain; Vitek MS identified this strain as *L. marthii*.

**Genomic analyses identified the *L. swaminathanii* FSL L7-0020^T draft genome contains genes associated with motility and antibiotic resistance, but lacked genes that confer (i) virulence, (ii) nitrate and nitrite reductase activity, and (iii) resistance to metal and sanitizers.** All 26 flagellar genes (Table S3) included in the reference database (BIGSdb-*Lm*) were detected, which correlates with the observation that *L. swaminathanii* FSL L7-0020^T is motile. The antimicrobial resistance genes *lin*, *fosX*, and *L. monocytogenes*'s *mprF* were detected following analyses using the Comprehensive Resistance Database (CARD 3.1.0) and the Resistance Gene Identifier (RGI 5.1.1) (46). For the virulence assessment, *L. swaminathanii* was initially assessed for the six virulence genes (*prfA*, *plcA*, *hly*, *mpl*, *actA*, and *plcB*) found on the *Listeria* Pathogenicity Island 1 (LIPI-1), and the internalin genes *inlA* and *inlB*; none of these were detected which supports this species is not pathogenic. Further analyses of *L. swaminathanii* draft genome did not identify any of the genes in the *Listeria* Pathogenicity Islands 2, 3, and 4 (LIPI-2, LIPI-3, LIPI-4). To further support that *L.*

**TABLE 1** Summary of the phenotypic characteristics of *L. swaminathanii* compared to previously reported characteristics of other species

| Characteristics[a] | Sensu stricto nov. Lsw[b] | Sensu stricto Lmo | Lma | Lin | Lws | Liv | Lse | Lcs | Lfr | Lim | Sensu lato Lgy | Lfc | Lgo | Lfl | Lth | Lva | Lco | Laq | Lny | Lcn | Lro | Lwp | Lgd | Lri | Lbo | Lpo | Lru |
|---|---|---|---|---|---|---|---|---|---|---|---|---|---|---|---|---|---|---|---|---|---|---|---|---|---|---|---|
| Voges-Proskauer | + | + | + | + | + | + | + | + | + | + | + | + | − | − | + | + | + | + | + | + | + | + | + | + | + | − | − |
| Nitrate reduction | − | − | − | − | − | − | − | − | − | + | V* | + | − | − | + | + | + | + | + | + | + | + | + | + | + | + | + |
| Motility | + | + | + | + | + | + | + | + | + | − | + | − | − | − | + | + | + | − | − | − | − | − | − | − | − | − | − |
| Hemolysis | − | + | − | − | + | + | + | − | + | − | + | − | +(α) | − | + | − | + | − | − | − | − | − | − | − | − | − | − |
| PI-PLC | + | + | − | − | + | + | − | − | + | − | + | − | − | − | − | − | − | − | − | − | − | − | − | − | − | − | − |
| D-Arylamidase | − | − | − | + | − | − | − | + | + | − | − | − | − | − | + | − | + | − | − | + | − | − | − | + | + | + | − |
| α-Mannosidase | + | + | + | + | + | + | + | + | + | + | V | − | − | − | + | + | + | + | + | + | + | + | + | + | + | + | + |
| D-Arabitol | + | + | + | + | + | + | − | − | − | + | + | + | + | + | + | + | + | + | − | + | + | + | + | + | + | (+) | (+) |
| D-Xylose | − | − | − | − | − | − | − | − | − | − | − | + | + | + | (+) | + | + | − | + | − | − | − | + | − | + | + | + |
| L-Rhamnose | − | + | − | + | + | − | + | + | − | + | V* | + | + | + | + | + | + | V | − | + | + | + | + | + | + | + | + |
| Methyl-α-D-Glucopyranoside | + | + | + | + | + | + | + | + | + | + | + | + | + | + | + | + | + | + | + | + | + | + | + | + | + | + | + |
| Methyl-α-D-Mannopyranoside | + | + | + | + | + | + | + | + | + | + | + | + | + | + | + | + | + | + | + | + | + | + | + | + | + | + | + |
| D-Ribose | − | − | − | − | V* | − | − | V† | − | + | + | + | + | + | + | + | + | + | + | + | + | + | + | + | + | + | + |
| Glucose-1-Phosphate | − | − | − | − | V | − | − | − | − | − | − | V! | − | − | − | − | − | − | − | − | − | − | − | − | V | − | − |
| D-Tagatose | − | − | − | + | + | + | + | − | − | − | − | − | − | − | + | V | + | + | − | − | − | − | − | − | − | − | − |
| Glycerol | + | V | − | + | + | + | + | + | + | V | V | + | (+) | + | (+) | + | V | V | + | V | + | + | + | V | + | + | + |
| L-Arabinose | − | − | − | − | − | − | + | − | − | − | − | − | − | + | + | + | + | V | V | + | − | − | + | + | + | + | + |
| D-Galactose | + | V | − | − | + | V | − | − | − | + | + | + | + | + | + | + | + | + | + | + | + | + | + | + | + | + | + |
| D-Glucose | + | V! | V! | V! | V! | + | + | + | + | + | V! | V! | + | + | + | + | + | + | + | + | + | + | + | + | + | + | + |
| L-Sorbose | − | V! | V! | V! | V! | + | + | + | − | − | V! | V! | + | + | + | + | + | − | + | + | + | + | + | + | + | + | + |
| Inositol | − | − | − | − | − | − | − | − | − | − | − | − | − | − | + | + | V | V | − | − | − | − | V | V | − | − | − |
| D-Mannitol | − | − | − | − | + | − | + | − | − | − | + | V | − | − | + | + | + | − | + | + | + | + | + | V | + | + | + |
| D-Maltose | + | + | + | + | + | + | + | + | + | V | + | + | + | + | + | + | + | − | + | + | + | + | + | + | + | + | + |
| D-Lactose | + | V! | + | + | V | + | + | + | + | V | + | V* | + | + | + | + | + | − | (+) | + | V! | − | + | + | V! | (+) | − |
| D-Melibiose | − | V! | − | V | V! | + | − | − | − | − | V | V | − | + | + | + | + | V | − | + | − | − | − | V | + | − | − |
| D-Sucrose | − | + | − | + | + | − | + | − | − | V | V | V | − | − | − | + | + | − | − | − | − | − | − | − | − | − | − |
| Inulin | − | V! | − | V! | − | − | − | − | − | − | − | − | − | − | − | − | − | − | − | − | − | − | − | − | − | − | − |
| D-Melezitose | − | V | V | V | V | V | − | − | − | V | − | V | − | − | − | − | − | − | − | − | − | − | − | − | − | − | − |
| D-Turanose | − | − | + | V | V | − | − | − | − | − | − | V* | + | + | − | − | + | − | − | − | − | − | − | − | − | − | − |
| D-Lyxose | − | V | V | V | V | − | − | − | − | − | V | V | − | − | − | V | − | V | − | − | − | − | − | − | − | − | − |

[a]Additional biochemical results from the API CH50 analysis can be found in Table S2. Consistent with all other currently described *Listeria* sp., *L. swaminathanii* hydrolyzes esculin and does not reduce nitrite.

[b]Lsw, *L. swaminathanii* (this study); Lmo, *L. monocytogenes* (1); Lma, *L. marthii* (6); Lin, *L. innocua* (3); Lws, *L. welshimeri* (5); Liv, *L. ivanovii* (4); Lse, *L. seeligeri* (5); Lcs, *L. cossartiae* (21); Lfr, *L. farberi* (21); Lim, *L. immobilis* (21); Lgy, *L. grayi* (2); Lfc, *L. fleischmannii* (8, 9); Lgo, *L. goaensis* (17); Lfl, *L. floridensis* (11); Lth, *L. thailandensis* (18); Lva, *L. valentina* (19); Lco, *L. costaricensis* (16); Laq, *L. aquatica* (11); Lny, *L. newyorkensis* (12); Lcn, *L. cornellensis* (11); Lro, *L. rocourtiae* (7); Lwp, *L. weihenstephanensis* (10); Lgd, *L. grandensis* (11); Lri, *L. riparia* (11); Lbo, *L. booriae* (12); Lpo, *L. portnoyi* (21); Lru, *L. rustica* (21).

[c]+, positive; (+), weak positive; V, variable; V!, variable between replicates and/or strains; V†, variable between studies; V‡, characteristic that differentiates subspecies; *L. ivanovii* subsp. *ivanovii* ferments ribose while subsp. *londoniensis* does not ferment ribose and subsp. *cayugensis* strains are variable for ribose fermentation; V*, *L. cossartiae* does not ferment ribose while subsp. *cossartiae* does not reduce nitrate and ferments methyl-α-D-glucopyranoside; *L. grayi* subsp. *grayi* does not reduce nitrate and ferments methyl-α-D-glucopyranoside, while subsp. *murrayi* reduces nitrate and does not ferment methy-α-D-glucopyranoside; *L. fleischmannii* subsp. *fleischmannii* ferments turanose, while subsp. *coloradonensis* does not ferment turanose; (α) alpha hemolysis observed; PI-PLC phosphatidylinositol-specific phospholipase C.

| *Listeria* genome | Virulence genes | | | Typical *Listeria* sensu stricto gene profile | | | | | Antimicrobial resistance | | | |
|---|---|---|---|---|---|---|---|---|---|---|---|---|
| | LIPI-1 | *inlA* | *inlB* | Flagella locus | *kat* | *sod* | Nitrate reductase | Nitrite reductase | *lin* | FosX | LM *mprF* | *nomB* |
| *L. monocytogenes* ATCC 19117 (lineage I) | ■ | ■ | ■ | ■ | ■ | ■ | | | ■ | ■ | ■ | |
| *L. monocytogenes* NCTC 10357ᵀ (lineage II) | ■ | ■ | ■ | ■ | ■ | ■ | | | ■ | ■ | ■ | ■ |
| *L. monocytogenes* SLCC 2479 (lineage II) | ■ | ■ | ■ | ■ | ■ | ■ | | | ■ | ■ | ■ | |
| *L. monocytogenes* SLCC 2376 (lineage III) | ■ | ■ | ■ | ■ | ■ | ■ | | | ■ | ■ | ■ | |
| *L. monocytogenes* FSL J1-0208 (lineage IV) | ■ | | | ■ | ■ | ■ | | | ■ | ■ | ■ | |
| *L. swaminathanii* FSL L7-0020ᵀ | | | | | ▨ | ■ | | | ■ | ■ | ■ | |

**FIG 3** Presence/absence of key genes from the *L. swaminathanii* FSL L7-0020ᵀ draft genome analyses compared to the *L. monocytogenes* genomes representing each of the four lineages and the type strain. Gray squares indicate the respective operon, loci, or gene (identified at the top of the column) is present, white indicates it is absent. For *kat* and *L. swaminathanii* FSL L7-0020ᵀ, the square is not a solid gray color to reflect this gene was detected, but may not be functional (i.e., this strain is catalase-negative). The flagella locus includes 26 genes (Table S1). The nitrate reductase results include genomic analyses for *narG*, *narH*, and *narI*. The nitrite reductase analyses include genomic analyses for (*nirB*, *nirD*).

*swaminathanii* does not reduce nitrate or nitrite, we also analyzed the draft genome for genes that encode nitrate (*narI*, *narH*, *narG*) and nitrite (*nirB*, *nirD*) reductases; these genes were not detected in the draft genome. Genes that were previously reported to confer reduced sensitivity to quaternary ammonium compounds (*qac*, *bcrABC*, *ermE*) were not detected nor were genes reported to confer resistance to cadmium (*cadA*, *cadC*). A visualization of the genomic results for *Listeria* key genes is presented in Fig. 3.

**Nonsynonymous substitutions were identified as the most likely cause for lack of catalase activity.** The *L. swaminathanii* FSL L7-0020ᵀ draft genome contained sequences that matched the entire catalase (*kat*; lmo2785, 1,467 bp) and superoxide dismutase (*sod*; lmo1439, 609 bp) gene sequences with no premature stop codons in either gene. Importantly, an analysis reported by Hudson at al. (45) compared the *kat* sequence for FSL L7-0020ᵀ with the *kat* sequences for one isolate of the closely related species *L. marthii* (UTK_C1-0015-E1) and two *L. swaminathanii* (UTK C1-0015 and UTK C1-0024) isolates that were obtained by this group; these three isolates tested catalase positive. This analysis identified four nonsynonymous nucleotide differences, with two consistent differences between the type strain and the three catalase-positive isolates. At amino acid positions 72 and 92, the *L. swaminathanii* type strain has glutamic acid (polar, acidic) and histidine (polar, basic), respectively, while the three catalase positive isolates have lysine (polar, basic) and arginine (polar, basic) at these sites. These amino acid differences may have an effect on the structure and function of the resulting protein, leading to the catalase-negative phenotype of FSL L7-0020ᵀ. Importantly, *L. marthii* UTK_C1-0015-E1 had the same amino acids as the two catalase-positive *L. swaminathanii* isolates, further supporting that conservation of these sites is important for catalase function. To better understand the impact of these amino acid changes, we investigated the frequency in which a glutamic acid is found at position 72 and histidine is found at position 92 using the multiple alignment of the catalase protein (12,367 sequences; https://pfam.xfam.org/family/PF00199#tabview=tab3) provided by Pfam, a database of Hidden Markov Models of protein families (47). Glutamic acid, the amino acid identified at position 72 in the *L. swaminathanii* FSL L7-0020ᵀ type strain occurs at a frequency of 5.2% while lysine (the amino acid at position 72 in the *kat* sequence in the three catalase positive isolates (detailed above) occurs at a frequency of 3.7%). Conversely, the histidine at position 92 has a frequency of < 0.01% (and only one of

the 12,367 sequences has a histidine in that position); the arginine at position 92 is highly conserved (93% frequency). These data suggest that an amino acid change from an arginine to a histidine at position 92 likely is responsible for the *L. swaminathanii* type strain specific loss of catalase.

**In silico PCR analysis identifies *L. swaminathanii* FSL L7-0020[T] as a *Listeria* species but does not assign a serotype.** A complete *prs* sequence was detected with no mismatches to either the forward or reverse primers, supporting that *L. swaminathanii* FSL L7-0020[T] would be identified as a *Listeria* species with the PCR assay described by Doumith et al. (48) The *L. swaminathanii* FSL L7-0020[T] genome yielded no BLAST hits for any of the four *L. monocytogenes* serovar specific sequences, indicating that *L. swaminathanii* FSL L7-0020[T] would only be identified as a *Listeria* spp. and would not be assigned to a serotype; hence the Doumith et al. (48) PCR assay would not misidentify *L. swaminathanii* FSL L7-0020[T] as *L. monocytogenes*.

**Description of *Listeria swaminathanii* sp. nov.** *L. swaminathanii* (swa.min.ath.anˉiˉi.i NL masc. adj. *swaminathanii* named in honor of Balasubramanian Swaminathan for his contributions to the epidemiology of human listeriosis and laboratory diagnostic methodologies).

*L. swaminathanii* FSL L7-0020[T] exhibits growth characteristics typical of nonpathogenic *sensu stricto Listeria* spp. except for the catalase reaction, the type strain for this species is catalase-negative. Gram-positive short rods. Oxidase negative. Facultative anaerobe. Presumed to be nonpathogenic based on the absence of hemolysis on SBA, lack of PI-PLC activity on LMCPM, and the absence of six virulence genes (*prfA*, *plcA*, *hly*, *mpl*, *actA*, and *plcB*) located on LIPI-1. Colonies on MOX are round, black, approximately 2 to 3 mm in diameter with a sunken center. Colonies on LMCPM were of similar size and shape as colonies on MOX and are opaque-white in color. Classic umbrella-patterned motility in MTM incubated at 25°C. Tumbling motility is observed microscopically at 25°C. Nonmotile at 37°C. Growth occurs between 4 and 41°C in BHI broth with optimal growth achieved between 30 and 37°C. The type strain does not reduce nitrate or nitrite. Phenotypically this type strain cannot be differentiated from *L. marthii* or *L. cossartiae* subsp. *cossartiae* using API *Listeria* (i.e., API numerical profile = 6110) or the biochemical reactions specified in the FDA BAM or ISO 11290-1:2017 methods. Voges-Proskauer positive. *L. swaminathanii* FSL L7-0020[T] is negative for D-arylamidase activity and positive for α-mannosidase activity. The type strain does not ferment D-xylose, L-rhamnose, D-ribose, glucose-1-phosphate, D-tagatose, L-arabinose, D-galactose, L-sorbose, inositol, D-mannitol, D-melibiose, D-sucrose, inulin, D-melezitose, D-turanose, or D-lyxose. Positive for fermentation of D-arabitol, methyl-a-D-glucopyranoside, methyl-a-D-mannopyranoside, glycerol, D-glucose, D-maltose, D-lactose, and starch. *L. swaminathanii* FSL L7-0020[T] is differentiated from *L. marthii* by the utilization of glycerol and lack of ability to ferment D-turanose. Differentiation from *L. cossartiae* subsp. *cossartiae* is achieved by the ability to utilize starch.

## DISCUSSION

**Three whole-genome sequence-based classification methods identified *L. swaminathanii* as a novel *Listeria sensu stricto* species; however, phenotypic-based differentiation from other species was challenging.** *L. swaminathanii* FSL L7-0020[T] WGS classification analyses confirmed placement of this species within the *Listeria* genus as a novel *sensu stricto* species based on meeting widely accepted species delineation thresholds (ANIb <95%, isDDH <70% [37]). All three WGS-based computational tools (ANIb, isDDH, and GTDB-Tk) used in this study to assess the *L. swaminathanii* FSL L7-0020[T] draft genome showed this species clusters closest to *L. marthii*. Phenotypically, *L. swaminathanii* FSL L7-0020[T] may be distinguished from other *sensu stricto* based on the unique catalase-negative attribute; however, this attribute could also lead to a situation where *L. swaminathanii* FSL L7-0020[T] isolates may not even be identified as *Listeria* as a catalase-positive reaction is often used to confirm *Listeria* to the genus level (38, 49, 50). Interestingly, several cases of human listeriosis have been attributed to catalase-negative *L. monocytogenes* strains (40–43), which supports a need to reduce the reliance on the catalase test for identification of *Listeria* spp. Recent characterization of additional *L. swaminathanii* isolates performed by

the University of Tennessee (29) (and reported after the initial submission of this work) shows that the catalase negative phenotype may be a strain-specific phenotype (similar to what has been reported for *L. monocytogenes*). Beyond the catalase test, phenotypically differentiating *L. swaminathanii* FSL L7-0020$^T$ from *L. marthii* and/or *L. cossartiae* subsp. *cossartiae* was difficult; these three species shared the same biochemical results for the species identification tests detailed in the reference methods (beta-hemolysis, rhamnose, xylose, mannitol) and generate the same numeric code with API *Listeria* (6110). The API CH50 glycerol, and D-turanose tests provided further species-level discrimination between *L. swaminathanii* FSL L7-0020$^T$ and *L. marthii*, and the starch test allowed for further differentiation of *L. swaminathanii* FSL L7-0020$^T$ from *L. cossartiae*.

**L. swaminathanii along with the other recently described Listeria sensu stricto species may not be detected using rapid detection methods and/or commonly used reference methods.** For many *Listeria* detection methods (both rapid and cultural), validation studies only included the "classical" six *Listeria* spp. (i.e., *L. monocytogenes*, *L. innocua*, *L. ivanovii*, *L. seeligeri*, *L. welshimeri*, and *L. grayi*), with some studies also validated with *L. marthii*. For a number of assays currently on the market it is hence unknown whether they detect the novel *Listeria* species identified since 2010. This lack of information was less of a concern until recently, given that until 2021 most newly described species (14 out of 15) were classified in the *sensu lato* clade and the food industry is more concerned with detecting *sensu stricto* species as this clade contains *L. monocytogenes* and the species most similar to *L. monocytogenes*. However, with the identification of *L. swaminathanii* and the recent publication of *L. cossartiae*, *L. farberi*, and *L. immobilis* (26), there are now 10 *Listeria sensu stricto* species, including four that were reported since 2021, which adds urgency to the need to evaluate existing methods for their ability to detect all *Listeria* sp. This need has also been recognized by the recent revision of the ISO reference method for *Listeria* spp. detection (ISO 11290-1;), which now includes the expected biochemical results for 11 of 20 recently described *Listeria* species (50). Hence, it is likely that future assay evaluation will include more of the recently described *Listeria* spp., particularly since the validation requirements for strain selection specified by AFNOR, AOAC, and MicroVal all state that the strain set selected for the inclusivity panels must reflect the diversity of the organisms being tested (51, 52).

Even if a novel *sensu stricto* species is detected by a rapid method (e.g., PCR) or yields *Listeria*-like colonies with cultural methods on the selective and differential agars, there is a strong potential for either misidentification or a false negative with the subsequent confirmatory tests. Currently, catalase and motility tests are utilized to confirm *Listeria* to the genus-level as catalase-positive and motility are considered universal traits to all *sensu stricto* species (13, 38, 49, 50); however, there is now sufficient evidence to warrant revising this claim. In addition to the potential for a catalase-negative *Listeria* with *L. swaminathanii* and some strains of *L. moncytogenes* as detailed above, the recently described *sensu stricto* species, *L. immobilis*, is nonmotile (26). After the genus-level confirmatory tests, the species-level tests also showed potential for a false negative or misidentification. While gaps between the reference methods and recent publications are expected due to the time required to update these methods, it is important to note that, presently, the reference methods do not list the expected results for the classic biochemical identification test (beta-hemolysis, rhamnose, xylose, and mannitol) for the recently described *sensu stricto* species. As an example, *L. swaminathanii* FSL L7-0020$^T$ is nonhemolytic, and negative for rhamnose, xylose, and mannitol fermentation, which is a profile not currently associated with any species in the commonly used reference methods (e.g., FDA BAM, Health Canada, ISO).

**The three rapid identification methods evaluated here showed a strong potential to misidentify the recently described Listeria sensu stricto species.** As rapid bacterial identification methods are becoming increasingly popular in the food industry, our data highlight the importance of updating reference databases to include at least all the *sensu stricto* species. Unlike the novel *sensu lato*, which historically do not generate acceptable species identifications with the rapid identification methods, the five recently described novel *sensu stricto* species (*L. cossartiae*, *L. farberi*, *L. immobilis*, *L. marthii*) along with the species reported

here (*L. swaminathanii*) were all misidentified with the rapid identification methods used in this study, including two biochemical methods (i.e., API Listeria, Vitek 2) and one MALDI-TOF method (i.e., Vitek MS). Notably, we saw the potential for a strain of a nonpathogenic novel *sensu stricto* species to be identified as the pathogenic species *L. monocytogenes* or *L. ivanovii*; at minimum, this could cause confusion and delays, and worse-case lead to unnecessary product disposals or recalls. Hence, future evaluation of different rapid identification methods (including methods not evaluated here, such as other MALDI -based methods (53) with larger *Listeria* strain sets as well as development of more inclusive databases will be important). Specifically, for the food industry, it will be important that the respective databases include strains representing all currently described *sensu stricto* species to ensure accurate identification. Importantly, these efforts (expansion of the identification databases) have already been initiated.

**L. swaminathanii should be recognized as a Listeria species despite not being able to achieve valid status.** In conclusion, while *L. swaminathanii* may not become a validly published species due to restrictions associated with its isolation from a US National Park, its designation as a novel *sensu stricto* species is firmly supported by the results described here. Incorporating this species, along with other recently described species, in *Listeria* method inclusivity studies will be important to ensure detection of all targeted *Listeria* species or to at least ensure that users have information as to which species are and are not detected with a given assay. This is important as a number of studies (21, 22, 24, 54) support the value of using *Listeria* spp., and particularly *Listeria sensu stricto* species, as index organism in environmental monitoring programs for food processing facilities. In addition, it will be important to include *L. swaminathanii* in studies that validate identification methods and in the reference databases for *Listeria* species identification methods, particularly since our data suggest that some systems, with their current databases, may misidentify *L. swaminathanii* as *L. monocytogenes*.

## MATERIALS AND METHODS

***Listeria* isolation and initial identification.** As part of a previously reported study evaluating the prevalence of *Listeria* in soil (25), a total of five soil samples were collected from the Great Smoky Mountains National Park and 25g aliquots of each sample were enriched in Buffered *Listeria* Enrichment Broth (BLEB, Becton, Dickinson, Frankland Lake, NJ, USA); *Listeria* spp. isolation was conducted as described in the US Food and Drug Administration's *Bacteriological Analytical Manual* (FDA BAM) Chapter 10 method (38) with one modification: Modified Oxford Agar (MOX, Becton, Dickinson) was incubated at 30°C instead of 35°C. From the options for selective and differential chromogenic agars detailed in FDA BAM, we used R&F *Listeria monocytogenes* Chromogenic plating medium (LMCPM, R&F Laboratories, Downers Grove, IL, USA). Following streaking of the five BLEB-enriched soil samples, the MOX and LMCPM agar plates were incubated for 48 h at 35°C, *Listeria*-like colonies were selected from both plate types and isolated onto Brain Heart Infusion (BHI, Beckton Dickinson) agar. Following isolation onto BHI, species identification was performed using a previously described protocol for PCR amplification and sequencing of the partial *sigB* gene (55).

**Whole-genome sequencing.** Genomic DNA was prepared and sequenced, using Illumina's MiSeq platform, as described in our previous publication (26). The raw sequencing data were assembled, and draft genome quality was assessed using the protocols described by Kovac et al. (56). Briefly, adapter sequences were trimmed using Trimmomatic 0.39 (57), and paired-end reads were assembled *de novo* using SPAdes v3.13.1 (58) with k-mer sizes of 33, 55, 77, 99, 127. Contigs <500 bp were removed, and assembly quality was checked using QUAST v5.0 (59), followed by screening for contamination using Kraken (60).

**Whole-genome-based phylogenic analysis.** Whole-genome sequence-based ANIb analysis was conducted on the *L. swaminathanii* FSL L7-0020$^T$ draft genome and a set of 34 reference genomes consisting of (i) the 30 type strains of all *Listeria* species and subspecies described as of May 17, 2021, and (ii) a representative for each of the four *L. monocytogenes* lineages (Fig. 1). Pyani (31) was used to calculate pairwise ANIb values, and a dendrogram was constructed using the dendextend R package (61). Further analysis by whole-genome sequence-based isDDH was also performed using the Genome-to-Genome Distance Calculator 2.1, formula 2 (identities/high-scoring segment pair [HSP]) (32). A newer WGS-based computational tool for classifying bacterial genomes, GTDB-Tk (released 2019, [35]), which is a software toolkit that classifies genomes using the Genome Taxonomy Database (GTDB; released in 2018 and updated biannually) (33), was also employed. GTDB infers phylogeny from a set of marker genes made up of 120 bacterial protein genes (bac120) (33), and GTDB-Tk assigns species classification based on ANI (calculated with FastANI), and Alignment Fraction (AF) (33). The same reference genomes used for ANIb were used for the GTDB-Tk analysis of *L. swaminathanii* FSL L7-0020$^T$. A phylogenetic tree was inferred from the GTDB-Tk output using RAxML (62), which utilized the alignment of the bac120 protein marker genes from all genomes assessed (the 34 reference genomes and the *L. swaminathanii*

draft genome) along with *Brochothrix thermosphacta* ATCC 11509$^T$ (output group). The tree was visualized using Figtree (63).

**Phenotypic analyses.** Phenotypic characterizations of *L. swaminathanii* FSL L7-0020$^T$ were carried out using BHI agar cultures streaked from a frozen stock culture (stored at −80°C in BHI broth supplemented with 15% glycerol), followed by incubation at 30°C for 24–36 h. Colony morphologies were assessed by streaking an overnight BHI broth culture onto selective and differential agars specified for *Listeria* isolation in the reference methods. Specifically, *L. swaminathanii* FSL L7-0020$^T$ was streaked to MOX and LMCPM agars, followed by incubation at 35°C for 48 h. *L. monocytogenes* 10403S and *L. innocua* ATCC 33090$^T$, were included as positive and negative controls, respectively.

Additional characterization tests performed included a combination of conventional tests outlined in commonly used reference methods for *Listeria* species identification (the FDA BAM Chapter 10 [38] and ISO 11290:2017 [50]), including (i) catalase, (ii) oxidase, (iii) Gram staining, (iv) beta-hemolysis on blood agar, (v) nitrate and nitrite reduction, and (vi) motility. Rapid biochemical test kits (i.e., API kits described below) were also used to perform certain standard classification tests (e.g., Voges-Proskauer, utilization of ribose, xylose, mannitol). Two biological replicates were performed for each test (including API tests). Catalase, oxidase, Gram-staining, and beta-hemolysis analyses were conducted as described in the reference methods (38, 50) using colonies grown on BHI agar as described above. *L. monocytogenes* 10403S and *L. booriae* FSL A5-0281$^T$ were included as negative and positive controls, respectively. Nitrate and nitrite reduction tests were performed in parallel using a method described by Buxton et al. (64). Briefly, a heavy inoculum from a freshly prepared BHI agar culture was inoculated into both Nitrite and Nitrate broths (prepared according to Buxton et al. [64]), followed by incubation at 35°C. Analyses were performed after 24 h and again after 5 days of incubation. Following incubation, aliquots of each culture were separately added to commercially prepared reagents of sulfanilic acid and N, N-dimethyl-*a*-nathylamine (commercially named NIT1 and NIT2, respectively, bioMérieux). When combined with NIT1 and NIT2, a red color change indicates the presence of nitrite in the nitrate enrichment broth (indicating that nitrate was reduced) or in the nitrite enrichment broth (indicating that nitrite was not reduced). Powdered zinc (bioMérieux), which reduces nitrate to nitrite, was added to the nitrate enrichments that did not exhibit a red color change. Following the addition of zinc, a red color change indicates nitrate was present; no color change indicates nitrate has been completely reduced to, nitric oxide, nitrous oxide, or molecular nitrogen (i.e., the species reduced nitrate).

Motility was assessed both microscopically and following stab inoculation into Motility Test Medium (MTM, Becton, Dickinson). For the microscopic method, wet mounts were prepared from BHI agar cultures grown at 25°C and 37°C for 24 h. Motility testing using MTM was performed by stab-inoculating the medium (purchased premade in 10 mL screw-cap tubes) with an isolated colony selected from BHI agar followed by incubation at 25°C with observations every 24 h for 7 days.

**Growth experiments.** We assessed growth of *L. swaminathanii* FSL L7-0020$^T$ at 4, 22, 30, 37, and 41°C by inoculating BHI broth with 30 to 300 CFU/mL, followed by incubation at the specified temperatures without shaking. The inoculum was verified by spread plating onto BHI agar followed by incubation for 24 to 36 h at 30°C. The BHI cultures incubated at 4°C were enumerated after 10 and 14 days, BHI cultures incubated at all other temperatures were enumerated after 24 and 48 h of incubation. *L. monocytogenes* 10403S was included as a positive control. Enumerations were carried out by serial diluting and spread plating 100 $\mu$L in duplicate onto BHI agar, followed by incubation at 30°C for 24 to 36 h. After incubation, colonies on BHI agar were counted using the automated SphereFlash colony counter (IUL Micro, Barcelona, Spain). Relative growth for each temperature was calculated as the average of the duplicate counts minus the starting inoculum. Anaerobic growth was assessed by streaking to BHI agar followed by incubation at 30°C for 24h under anaerobic conditions. The growth experiments were performed in two biological replicates.

**API *Listeria*, CH50, and 20E test kit analyses.** The API kit tests were performed per the manufacturer's instruction. Specifically, the API *Listeria* strips were prepared and incubated at 35°C for 18 to 24h. For API CH50, *L. swaminathanii* FSL L7-0020$^T$ was suspended in CHB/E medium, and the strips were inoculated per the manufacture's instruction, followed by aerobic incubation at 30°C for 48 h (reactions that were positive at this time point were considered positive). The API 20E was utilized because it includes tests classically used to characterize *Listeria* spp. to the genus level, including (i) Voges-Proskauer, (ii) indole, (iii) urease, and (iv) H$_2$S production. For API 20E, *L. swaminathanii* was suspended in NaCl 0.5% Medium (bioMérieux), and the strip was inoculated per the manufacturer's instruction, followed by incubation at 35°C for 24 h. API testing was performed in two biological replicates.

**Vitek 2 and Vitek MS analyses.** *L. swaminathanii* FSL L7-0020$^T$ along with the type strains for the recently reported novel *sensu stricto* species *L. cossartiae* (subsp. *cossartiae* FSL L7-1447$^T$ and subsp. *cayugensis* FSL L7-0993$^T$), *L. farberi* FSL L7-0091$^T$, and *L. immobilis* FSL L7-1519$^T$ were prepared and processed on the Vitek 2 (bioMérieux) V7.01 and Vitek MS (bioMérieux) V3.2 automated identification systems per the manufacturer's instructions. After inoculating the GP (Gram-Positive) reagent card, the Vitek-2 system automatically assesses 64 biochemical reactions that are compared to a database to generate an identification (65). Vitek MS is a Matrix-Assisted Laser Desorption Ionization Time-Of-Flight (MALDI-TOF) mass spectrometry method that automatically compares an isolate's spectrum to a database (66). For Vitek 2, unknown biochemical patterns are reported as outside the scope of the database. For Vitek MS, the resulting spectra are assigned a percent probability ranging from 60 to 99.9%. Values <60% are assigned to spectra too different from any in the database, such that no possible identification is provided.

***Listeria* catalase, virulence, flagellar, metal, sanitizer resistance, and nitrate and nitrite reductase gene analyses.** The nucleotide sequences for the catalase, virulence, flagella, and sanitizer resistance genes were downloaded from the open-access Institut Pasteur database, BIGSdb-*Lm*, as described by Moura et al. (67) and Ragon et al. (68). Reference sequences for the nitrate and nitrite reductase genes were obtained from NCBI. For nitrate reductase, we downloaded the sequences for the alpha, beta and gamma subunits (*narI, narH, narG*) and for nitrite reductase we downloaded the sequences for the small and large subunits (*nirB, nirD*) from the annotated *L. booriae* genome (NZ_JNFA01000024.1). Using the reference sequences, a BLASTn query of the *L. swaminathanii* FSL L7-0020$^T$ draft genome was performed.

**Antimicrobial resistance gene analysis.** The *L. swaminathanii* FSL L7-0200$^T$ draft genome was analyzed for genes that confer antimicrobial resistance using the Comprehensive Antibiotic Resistance Database (CARD 3.2.0) (46). The draft genome was uploaded to the Resistance Gene Identifier (RGI 5.2.1) for analysis.

**Genomic investigation of the catalase-negative *L. swaminathanii* FSL L7-0020$^T$ phenotype.** The *kat* and *sod* sequences of *L. swaminathanii* FSL L7-0020$^T$, *L. swaminathanii* UTK C1-0015, *L. swaminathanii* UTK C1-0024, and *L. marthii* UTK_C1-0015-E1 were aligned and searched for premature stop codons and variable sites using MEGA (69). For *kat*, the frequency of the identified amino acid changes specific to FSL L7-0020$^T$ were visualized using the Pfam full alignment of the catalase protein.

***In silico* PCR for *Listeria monocytogenes* serovars.** *In silico* PCR (isPCR) using the primers described by Doumith et al. (48) for differentiation of *L. monocytogenes* serovars was also performed. We first performed a BLASTn query of the *L. swaminathanii* draft genome against the *L. monocytogenes* serovar specific genes (*lmo0737, lmo118*, ORF2819, ORF2110) and a gene common to all described *Listeria* species (*prs*). The reference sequences were obtained from the BIGSdb-*Lm* database described above from the PCR Serogroup scheme. The sequences obtained from the query were subsequently tested against the primer sequences for an isPCR analysis.

**Data availability.** The draft genome total length is 2.8 Mb with a GC content of 38.7%. The type strain, FSL L7-0020$^T$, ATCC TSD-239$^T$ was isolated from soil collected in the Great Smoky Mountain National Park, NC, USA, on November 2, 2017.

The GenBank/EMBL/DDBJ accession numbers for the 16S rRNA and draft genome sequences for the type strain are MT117895 and JAATOD000000000, respectively.

## SUPPLEMENTAL MATERIAL

Supplemental material is available online only.

**SUPPLEMENTAL FILE 1**, PDF file, 0.4 MB.

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
