## [Reviewer comments · Microbiology Spectrum]

Microbiology Spectrum

Soil collected in the Great Smoky Mountains National Park yielded a novel *Listeria sensu stricto* species, *L. swaminathanii*

Catharine Carlin, Jingqiu Liao, Lauren Hudson, Tracey Peters, Thomas Denes, Renato Orsi, Xiaodong Guo, and Martin Wiedmann

Corresponding Author(s): Martin Wiedmann, Cornell University

Review Timeline:

Submission Date:	February 5, 2022
Editorial Decision:	March 4, 2022
Revision Received:	May 1, 2022
Accepted:	May 7, 2022

Editor: Benjamin Wolfe

Reviewer(s): Disclosure of reviewer identity is with reference to reviewer comments included in decision letter(s). The following individuals involved in review of your submission have agreed to reveal their identity: Alexandre Leclercq (Reviewer #1)

Transaction Report:

DOI: <https://doi.org/10.1128/spectrum.00442-22>

March 4, 2022

Dr. Martin Wiedmann
Cornell University
Department of Food Science
347 Stocking Hall
Ithaca, New York 14853

Re: Spectrum00442-22 (Soil collected in the Great Smoky Mountains National Park yielded a novel *Listeria* species, *L. swaminathanii*, effectively expanding the sensu stricto clade to ten species)

Dear Dr. Martin Wiedmann:

Thank you for submitting your manuscript to Microbiology Spectrum. Two experts have reviewed your manuscript and have provided feedback to improve clarity and add additional detail (please see comments below). I invite you to submit a revised manuscript that incorporates this feedback.

Link Not Available

Sincerely,

Benjamin Wolfe

Reviewer comments:

Reviewer #1 (Comments for the Author):

Dear Colleagues

This manuscript describes an interesting and well-described new species of *Listeria* of the so-called « sensu stricto » clade: *Listeria Swaminathanii*, isolated in the soil of the Great Smoky Mountains national park. I really support the publication of this new species.

GENERAL COMMENTS:

1/ Of the taxonomic point of view the species "swaminathanii" is receivable and has been already attributed to a species of Orchidaceae in India.

2/ Please revise the introduction of the manuscript to avoid any critic or assimilate critics to National Park. In two sentences, please sum up the problem. The readers could be informed but it is an official decision that could not be discussed in a scientific journal.

3/ L77-79 The reviewer doesn't understand this argument for not recognition of this new species or publication in IJSEM because the ATCC collection already sells the *Listeria swaminathanii* TSD-239 without restrictions except MTA. It is possible to have it so for the scientific.

4/ Please provide antimicrobial susceptibility results for this new species.

SPECIFIC COMMENTS:

Title. Please delete "effectively expanding... species" like this the title could not become false tomorrow and is out of time

Introduction: L49-L75, it is too long and the readers becomes lost. Please revise to see the impact to be in sensu stricto or sensu lato clade; explain that sensu stricto species have been used to establish the food safety called *Listeria* spp. and the problem to expand this clade but also the problem of new described species.

L108 confirmed or identified? In a taxonomic study it is more identified. See L137 the authors said, "identify"

L110 31 colonies from X? Petri plates to see the clonality

L125 add in brackets NCBI accession number

L132, L318, L414 it is ANIb

L138 similar: please provide in brackets the value

L142 please use *Listeria* sp. and not spp.

L149 put "generally detected by.."

L154 As these authors insist on the fact that this species belonged to sensu stricto clade, a question could be : this new species has same colonies on chromogenic or not media than other sensu stricto clade species? Please indicate something

L163 it is interesting that *costaricensis* and *swaminathanii* shared lack of catalase and GTDB-Tk output clusterized them together. Please i

L166 Please investigate the superoxide dismutase also. For *kat* gene, it is the previously reported mutations or not?

L171 read alone this sentence could say that only *Lm* is haemolytic. No, it is only your control *Lm*. Please precise to avoid any misinterpretation.

L172 and for the CAMP test?

L181 do these authors verify nitrate/nitrite reduction and motility by genome analysis too

L204 it is API-*Listeria* because it is a trade name and could not be changed

L218 These authors don't use the correct *L. innocua* strains that bioMérieux asks in its quality control that are at the limit positive for some API strip test that allowed to validate their accuracy for identification. Please improve.

L217 it is interesting that these authors not report some results on their API50CH

L221 API20E identified a *Listeria* ? These authors mean that some biochemical characteristics in API20E identified this species? Please improve.

L245 May you provide in supplemental method the reference spectra for this species for Maldi-tof MS. Please also use Bruker system that contains more new species (18 species) to improve this part. Using full extraction of proteins could change the results also.

L273 and other islands? LIPI-2,3,4? What is the genoserotype of this strain in the Doumith et al., 2004 scheme as it is explained isPCR L509?

L369 the lack of catalase could be due to stress action on the strain see relevant articles about *Lm* catalase negative.

L340 it is not true as i.e. Afnor certification validates commercial methods now with a set of *Listeria* strains including new species.

L351 "all *Listeria* spp." not accurate it is "all *Listeria* sp."

L366 true but remind that this method is not continuously updated so it is normal and not a drawback as these authors mentioned. ISO 11290 has been validated and published in 2017. Please modulate your sentences.

L367 These authors didn't test the two main system of Maldi-Tof MS to conclude like that.

L384 and 387. There's no scientific proof that there's a correlation between *L. swaminathanii* presence and presence of *Lm* as this new species could become a bioindicator of the potential presence of low amount of *Lm*. Please revise the last sentence of the conclusion or perform experiments.

L399 incubation temperature?

L433 Not all the tests described in BAM ISO are in your text, for not described test used in your manuscript, please provide a description.

L482 according to bioMérieux manual it is read as *Bacillus* at 48h, isn't it?

L508 Where is the metal resistance? It is a disinfectant tolerance not resistance, see Carpentier and Cerf article in IJFM on this distinction for *Lm*.

Table 1 please add nitrite reduction

Reviewer #2 (Comments for the Author):

In this manuscript, Carlin et al reported the use of whole-genome sequence-based average nucleotide identity BLAST and in

silico DNA-DNA Hybridization analyses to identify a new bacterial isolate collected from soil samples in Smoky Mountains as a novel *Listeria* species *L. swaminathanii*, with the highest similarity to a known species *L. marthii*. Some additional phenotypic and molecular tests were conducted on this new isolate including colony morphology, biochemical test, motility test, etc, to further confirm that this isolate belongs to *Listeria sensu stricto*. Overall, the experiments and data analysis were comprehensively carried out. This manuscript also provides a very interesting reading regarding to the current status of this and many other putatively novel bacterial species that cannot be validly published due to the restrictions and rules set by different parties and organizations. Minor comments: some additional discussion on the potential impact of this and other recently identified *Listeria* spp (carrying pathogenicity genes or not) on food safety and public health would be valuable. A figure that shows/compares the genomic mapping of this new species with *L. monocytogenes* would be helpful.

Staff Comments:

Preparing Revision Guidelines

Please return the manuscript within 60 days; if you cannot complete the modification within this time period, please contact me. If you do not wish to modify the manuscript and prefer to submit it to another journal, please notify me of your decision immediately so that the manuscript may be formally withdrawn from consideration by Microbiology Spectrum.

Response to Reviewer #1:

Comment: This manuscript describes an interesting and well-described new species of *Listeria* of the so-called « sensu stricto » clade: *Listeria Swaminathanii*, isolated in the soil of the Great Smoky Mountains national park. I really support the publication of this new species.

Response: Thank you. No response needed.

GENERAL COMMENTS:

Comment: 1/ Of the taxonomic point of view the species "swaminathanii" is receivable and has been already attributed to a species of Orchidaceae in India.

Response: We propose to retain the name "*Listeria swaminathanii*" as it is acceptable for species names in different genera to share the same name. An example already exists within *Listeria* where a species name is shared with another genus – *Listeria weihenstephanensis* and *Bacillus weihenstephanensis*.

Comment: 2/ Please revise the introduction of the manuscript to avoid any critic or assimilate critics to National Park. In two sentences, please sum up the problem. The readers could be informed but it is an official decision that could not be discussed in a scientific journal.

Response: The introduction has been revised to appear less critical of the National Park system; however, others have appreciated the details provided in the Introduction (e.g., reviewer 2) and we have thus tried to retain key information to provide guidance to others regarding how to publish species where type strains were only obtained from National Parks.

Comment: 3/ L77-79 The reviewer doesn't understand this argument for not recognition of this new species or publication in IJSEM because the ATCC collection already sells the *Listeria swaminathanii* TSD-239 without restrictions except MTA. It is possible to have it so for the scientific.

Response: The *L. swaminathanii* is deposited into a special National Park collection within ATCC, which requires an additional MTA specifically for the National Park. A recent decision (October 2020) by the International Committee on Systematics of Prokaryotes (ICSP) has deemed the National Park MTA too restrictive. Details of this discussion can be found here: <https://www.the-icsp.org/reports>; an excerpt from the meeting minutes states "*The MTA from the US National Parks Authority attached to the type strain is not acceptable for type strains as it is considered to contravene Rule 30(4) of the ICNP,*". Therefore, while *L. swaminathanii* is deposited in ATCC, ICSP does not recognize this deposition as a valid deposition for description of a new species (which requires the type strain to be deposited into two collections that allow for open access) due to the restrictive National Park MTA that needs be signed to obtain this strain. We have clarified this in lines 89-98.

Comment: 4/ Please provide antimicrobial susceptibility results for this new species.

Response: An evaluation of antimicrobial susceptibility has been performed and the results were added to the manuscript (lines 283-290).

SPECIFIC COMMENTS:

Comment: Title. Please delete "effectively expanding... species" like this the title could not become false tomorrow and is out of time

Response: The title has been revised.

Comment: Introduction: L49-L75, it is too long and the readers becomes lost. Please revise to see the impact to be in sensu stricto or sensu lato clade; explain that sensu stricto species have been used to establish the food safety called *Listeria* spp. and the problem to expand this clade but also the problem of new described species.

Response: The introduction has been revised.

Comment: L108 confirmed or identified? In a taxonomic study it is more identified. See L137 the authors said, "identify"

Response: The manuscript has been revised to say "identified" (line 111).

Comment: L110 31 colonies from X? Petri plates to see the clonality

Response: This section has been revised to clarify colony source (line 114).

Comment: L125 add in brackets NCBI accession number

Response: The NCBI accession number has been added (line 129).

Comment: L132, L318, L414 it is ANIb

Response: This change has been made throughout the manuscript.

Comment: L138 similar: please provide in brackets the value

Response: This change has been made (line 142).

Comment: L142 please use *Listeria* sp. and not spp.

Response: This change has been made (line 146).

Comment: L149 put "generally detected by.."

Response: We revised the statement as requested (line 155).

Comment: L154 As these authors insist on the fact that this species belonged to sensu stricto clade, a question could be: this new species has same colonies on chromogenic or not media than other sensu stricto clade species? Please indicate something

Response: This section was revised to clarify that the *L. swaminathanii* colony morphology is indistinguishable from other non-pathogenic *Listeria sensu stricto* species on chromogenic media (lines 148-150).

Comment: L163 it is interesting that *costaricensis* and *swaminathanii* shared lack of catalase and GTDB-Tk output clusterized them together. Please i

Response: It appears the reviewer has confused *L. costaricensis* for *L. cossartiae*. *L. costaricensis* does lack catalase activity, but it is a *sensu lato Listeria* that does not cluster close to *L. swaminathanii*. *L. cossartiae* does cluster close to *L. swaminathanii*, however, it is catalase-positive.

Comment: L166 Please investigate the superoxide dismutase also. For kat gene, it is the previously reported mutations or not?

Response: We updated the result section to indicate that we identified a nonsynonymous mutation in the kat gene that can likely explain the catalase negative phenotype; this mutation was not reported before and is located in a highly conserved aa site (aa 92 in *kat*). We also compared the *sod* gene sequences between the catalase-negative *L. swaminathanii* reported here and two catalase-positive *L. swaminathanii* recently reported by Hudson et al. and identified no non-synonymous changes (or frame shift mutations) between the catalase positive and negative *L. swaminathanii*. This has been added to the revised manuscript (lines 302-331).

Comment: L171 read alone this sentence could say that only Lm is haemolytic. No, it is only your control Lm. Please precise to avoid any misinterpretation.

Response: We revised this sentence to state that *L. monocytogenes* 10403S was hemolytic (line 182).

Comment: L172 and for the CAMP test?

Response: Instead of using CAMP to support the absence of hemolysis observed for *L. swaminathanii*, we analyzed the genome for the hemolysin gene (*hly*). Additional details were added to the manuscript clarifying that the phenotypic lack of hemolysis is supported by the absence of *hly* from the draft genome (lines 184-185). Additionally, the CAMP test was not performed as a number of publications support the potential for misinterpretation of CAMP test results (McKellar, 1994; Rodriguez et al., 1986, Skalka et al., 1983; Vazques et al., 1992).

Comment: L181 do these authors verify nitrate/nitrite reduction and motility by genome analysis too

Response: The draft genome was analyzed for motility genes and the manuscript was revised to clarify the genome analysis (flagella locus present) correlated with the motility observations (line 287). Genomic analyses for genes encoding nitrate and nitrite reductases were also performed and no evidence was found for presence of genes encoding nitrate and nitrite reductases; these results were also added to the manuscript (lines 295-298).

Comment: L204 it is API-Listeria because it is a trade name and could not be changed

Response: This change was made throughout the manuscript.

Comment: L218 These authors don't use the correct *L. innocua* strains that bioMérieux asks in its quality control that are at the limit positive for some API strip test that allowed to validate their accuracy for identification. Please improve.

Response: bioMérieux states to use *L. innocua* ATCC 33090 to test for degradation of the DIM and XYL test reagents; this has been clarified in the manuscript. This is the strain that we used (lines 233-234).

Comment: L217 it is interesting that these authors not report some results on their API50CH

Response: All API CH50 results are reported in Supplemental Table S2. A footnote has been added to Table 1 to clarify this.

Comment: L221 API20E identified a *Listeria*? These authors mean that some biochemical characteristics in API20E identified this species? Please improve.

Response: The header and text of this section have been revised to clarify that specific reactions included in the API 20E (i.e., Voges Proskauer) can provide relevant results for isolate characterization (lines 237-241).

Comment: L245 May you provide in supplemental method the reference spectra for this species for Maldi-tof MS. Please also use Bruker system that contains more new species (18 species) to improve this part. Using full extraction of proteins could change the results also.

Response: We have added a new supplemental Figure S1, which provides reference spectra for *L. swaminathanii* for VITEK MS MALDI TOF. We also revised the “Discussion” section to clarify that future evaluation of different rapid identification methods (including methods not evaluated here, such as the Bruker MS system]) with larger *Listeria* strain sets as well as development of more inclusive databases will be important. We have also clarified that some of these efforts have already been initiated, for example for the Bruker system where the database has been extended to contain more new species (lines 411-417).

Comment: L273 and other islands? LIPI-2,3,4? What is the genoserotype of this strain in the Doumith et al., 2004 scheme as it is explained isPCR L509?

Response: We have revised the results section to clarify that (i) a complete prs sequence was detected with no mismatches to either the forward or reverse primers supporting that *L. swaminathanii* FSL L7-0020 would be identified as a *Listeria* and that (ii) the *L. swaminathanii* FSL L7-0020 genome yielded no BLAST hits for any of the *L. monocytogenes* serovar specific sequences. This indicates that *L. swaminathanii* would only be identified as a *Listeria* spp. and would not be assigned to a serotype; hence the Doumith et al PCR assay (Doumith et al., 2004) would not misidentify *L. swaminathanii* FSL L7-0020^T as *L. monocytogenes* (lines 337-340).

We also revised the manuscript to clarify that none of the genes in LIPI-2,3,4 were identified on the *L. swaminathanii* FSL L7-0020^T genome (lines 293-295).

Comment: L369 the lack of catalase could be due to stress action on the strain see relevant articles about Lm catalase negative.

Response: We revised the manuscript to clarify that we subcultured *L. swaminathanii* FSL L7-0020^T and that it maintained its catalase negative phenotype; this is important as a previous publication (Bubert et al., 1997) indicated that upon subculturing, one out of two catalase negative *L. monocytogenes* isolates reverted to a catalase positive phenotype (lines 169-176). We assume that this is the article this reviewer referred to. Combined with identification of a nonsynonymous change in a conserved aa in *kat* this

suggests that the catalase negative phenotype for *L. swaminathanii* FSL L7-0020^T is nonreversible; this has also been clarified in the revised manuscript.

Comment: L340 it is not true as i.e. Afnor certification validates commercial methods now with a set of *Listeria* strains including new species.

Response: We revised this section (lines 411-417) to clarify that recent changes will increase the likelihood that assays evaluated now and in the future will be validated with a larger number of new *Listeria* species. For example, the recent revision of the ISO *Listeria* species reference method (ISO 11290-1) now includes the expected biochemical results for 11 of 20 recently described *Listeria* species and hence, it is likely that future assay evaluation will include more of the recently described *Listeria* spp., particularly since the validation requirements for strain selection specified by AFNOR, AOAC, and MicroVal all state the strain set selected for the inclusivity panels must reflect the diversity of the organisms being tested. It is important to note however that we were not able to locate a reference for an established panel of strains to be used for method validations and that a certificate granted in 2021 by AFNOR for a *Listeria* spp. media (certificate # NEO 35/05-07/16) only included a strain set consisting of six species with no recently described species included.

Comment: L351 "all *Listeria* spp." not accurate it is "all *Listeria* sp."

Response: This change has been made (line 411).

Comment: L366 true but remind that this method is not continuously updated so it is normal and not a drawback as these authors mentioned. ISO 11290 has been validated and published in 2017. Please modulate your sentences.

Response: We revised the text to clarify that it is expected that updates to references methods require time (lines 427-431).

Comment: L367 These authors didn't test the two main system of Maldi-Tof MS to conclude like that.

Response: We revised this section to clarify that future additional evaluation of other MALDI-TOF MS systems is needed (lines 447-449).

Comment: L384 and 387. There's no scientific proof that there's a correlation between *L. swaminathanii* presence and presence of Lm as this new species could become a bioindicator of the potential presence of low amount of Lm. Please revise the last sentence of the conclusion or perform experiments.

Response: We have completely revised this section to remove any statements that could be interpreted as suggesting that *L. swaminathanii* per se is an index organism for *L. monocytogenes*.

Comment: L399 incubation temperature?

Response: The incubation temperature has been added (line 524).

Comment: L433 Not all the tests described in BAM ISO are in your text, for not described test used in your manuscript, please provide a description.

Response: We have revised and re-organized this section to better describe all tests used (lines 514-520).

Comment: L482 according to bioMérieux manual it is read as Bacillus at 48h, isn't it?

Response: We have simplified this section to clarify that 48 h results were recorded and used (lines 559-561).

Comment: L508 Where is the metal resistance? It is a disinfectant tolerance not resistance, see Carpentier and Cerf article in IJFM on this distinction for Lm.

Response: We analyzed the draft genome for genes that confer metal resistance. The results (no metal resistance genes detected) have been added to the manuscript (lines 299-300).

Comment: Table 1 please add nitrite reduction

Response: A footnote was added to Table 1 to indicate that all currently described *Listeria* are negative for nitrite reduction.

Response to Reviewer #2:

Comment: In this manuscript, Carlin et al reported the use of whole-genome sequence-based average nucleotide identity BLAST and in silico DNA-DNA Hybridization analyses to identify a new bacterial isolate collected from soil samples in Smoky Mountains as a novel *Listeria* species *L. swaminathanii*, with the highest similarity to a known species *L. marthii*. Some additional phenotypic and molecular tests were conducted on this new isolate including colony morphology, biochemical test, motility test, etc, to further confirm that this isolate belongs to *Listeria sensu stricto*. Overall, the experiments and data analysis were comprehensively carried out. This manuscript also provides a very interesting reading regarding to the current status of this and many other putatively novel bacterial species that cannot be validly published due to the restrictions and rules set by different parties and organizations. Minor comments: some additional discussion on the potential impact of this and other recently identified *Listeria* spp. (carrying pathogenicity genes or not) on food safety and public health would be valuable. A figure that shows/compares the genomic mapping of this new species with *L. monocytogenes* would be helpful.

Response: We have revised the Importance, Introduction, and Discussion section and particularly the last section of the discussion entitled "*L. swaminathanii* should be recognized as a *Listeria* species despite not being able to achieve valid status" to further clarify the potential impact of this and other recently identified *Listeria* spp. (carrying pathogenicity genes or not) on food safety and public health, such as the importance of including these new species in evaluation of detection and identification methods. A figure has been created (Fig. 3) showing key results from the genomic comparison of *L. swaminathanii* compared to *L. monocytogenes*.

May 7, 2022

Dr. Martin Wiedmann
Cornell University
Department of Food Science
347 Stocking Hall
Ithaca, New York 14853

Re: Spectrum00442-22R1 (Soil collected in the Great Smoky Mountains National Park yielded a novel *Listeria sensu stricto* species, *L. swaminathanii*)

Dear Dr. Martin Wiedmann:

Thank you for making very thorough and thoughtful revisions to your manuscript. I am pleased to inform you that your manuscript has been accepted for publication. I am forwarding it to the ASM Journals Department. You will be notified when your proofs are ready to be viewed.

Sincerely,

Benjamin Wolfe
Editor, Microbiology Spectrum

Journals Department
Supplemental Material for Publication: Accept